# Pivot Pre-finetuning for Low Resource MT: A Case Study in Kikamba

**Stephen Kiilu**
AMMI-AIMS Senegal
skiilu@aimsammi.org

**Machel Reid**
Google Research, Brain Team
machelreid@google.com

## Abstract

Current approaches to performant machine translation often require large amounts of data (Koehn et al., 2022). However, a majority of the 7,000+ languages in the world often have a relative lack of digitized/organized text available and are considered low-resource. In practical terms, this often means that there is a substantial drop in quality between translation performance between high and low-resource language pairs. We look to explore the intersection of rapid NMT adaptation techniques and pre-trained sequence-to-sequence models to better leverage multilingual models, performing a case study on Kikamba.

## 1 Using pivot languages for extremely low-resource machine translation fine-tuning

In this paper, we look to investigate the usage of pivot languages (i.e. intermediate languages) to bridge the gap from a pre-trained multilingual sequence-to-sequence model (e.g. mT5: Xue et al., 2021; mBART: Liu et al., 2020) and extremely low-resource machine translation (e.g. English to Kikamba). Particularly, we follow the approach of Neubig & Hu (2018), which proposed a related language-pair pre-training approach for rapid adaptation of NMT systems to low-resource languages. The availability of multilingual pre-trained language models (e.g. XLM-R, mBART, mT5) has lessened the necessity of this, and in some cases, have even improved over the related-language translation pre-training approach (Reid et al., 2021) for low-resource languages with ≥100k pairs.

However, in the case of extremely low-resource languages, with limited high-quality and still limited noisily aligned pairs, we propose to leverage the approach of Neubig & Hu (2018) with pre-trained language models in the loop. This takes advantage of the data-efficiency of pre-trained LMs, while also leveraging Neubig & Hu (2018)'s rapid NMT approach without having to pre-train a language model with machine translation data from scratch for increased performance (Reid & Artetxe, 2022). We refer to this approach as *pivot pre-finetuning*, as we leverage a pivot language pair for a pre-finetuning procedure to enable data efficient machine translation performance.

## 2 Experimental Setup & Results

To validate the use of pivot pre-finetuning, we compare direct Kikamba fine-tuning of a pre-trained massively multilingual sequence-to-sequence model (mT5; Xue et al., 2021) and various intermediate pre-finetuning languages spanning different resource levels and dataset sizes.

### 2.1 Experimental Setup

**Model** We choose the 300M parameter mT5-small (Xue et al., 2021) to be the backbone for our experiments. Leveraging pre-trained multilingual sequence-to-sequence models for machine translation has shown not only improved performance but also improved data-efficiency (Liu et al., 2020).

**Datasets** We use three pivot languages: Kikuyu (another extremely low-resource language, but most similar to Kikamba), Kiswahili/Kinyarwanda (relatively mid-resourced languages, however more distant than Kikuyu), and French (an extremely high-resource European language). For each

| Target | Pivot lang | Resource-level | # Examples | Backbone | BLEU | ChrF |
|--------|-----------|---------------|-----------|----------|------|------|
| Kikamba | None | - | 0.5k | mT5-small | **0.0858** | **4.429** |
| Kikamba | Kikuyu | low | 1k | mT5-small | 0.0034 | 3.261 |
| Kikamba | Kinywaranda | mid | 1k | mT5-small | 0.0035 | 4.394 |
| Kikamba | Kiswahili | mid | 1k | mT5-small | 0.0065 | 3.906 |
| Kikamba | French | high | 1k | mT5-small | 0.0022 | 4.212 |
| Kikamba | None | - | 25k | mT5-small | 0.1296 | 7.805 |
| Kikamba | Kikuyu | low | 50k | mT5-small | 0.4662 | 11.143 |
| Kikamba | Kinywaranda | mid | 50k | mT5-small | 0.6487 | 10.735 |
| Kikamba | Kiswahili | mid | 50k | mT5-small | **0.7806** | **11.151** |
| Kikamba | French | high | 50k | mT5-small | 0.5730 | 10.665 |
| Kikamba | None | - | 50k | mT5-small | 0.0243 | 7.982 |
| Kikamba | Kikuyu | low | 100k | mT5-small | 0.086 | **11.625** |
| Kikamba | Kinywaranda | mid | 100k | mT5-small | 0.2823 | 10.993 |
| Kikamba | Kiswahili | mid | 100k | mT5-small | **0.3517** | 11.042 |
| Kikamba | French | high | 100k | mT5-small | 0.1705 | 9.798 |

Table 1: Comparison on various pivot pre-finetuning settings for Kikamba translations.

| Target | Pivot lang | # Examples | Backbone | # 1-star | # 2-star | # 3-star |
|--------|-----------|-----------|----------|----------|----------|----------|
| Kikamba | – | 0.5k | mT5-small | 100 | 0 | 0 |
| Kikamba | – | 25k | mT5-small | 100 | 0 | 0 |
| Kikamba | Kiswahili | 50k | mT5-small | 80 | 20 | 0 |
| Kikamba | Kiswahili | 100k | mT5-small | 80 | 19 | 1 |

Table 2: Human evaluation results. We sample 100 Kikamba sentences and perform human evaluation. 1-star means almost no fluency (only $< 30\%$ of the concepts are translated, 2-star – some fluency in the translation (there is context and about half of the concepts are translated) and 3-star means almost fluency (about $70\%$ correct translation).

language, we compare using varying amounts of *pivot pre-finetuning* and direct fine-tuning data pairs for training. We include details on training datasets in the Appendix.

**Evaluation** For evaluation, we use a subset of 500 pairs from the FLoRes-200 English-Kikamba `devtest` data. We evaluate using SacreBLEU (Post, 2018; Papineni et al., 2002) and chrF (Popović, 2015) given the morphologically-rich nature of Kikamba. Finally, we conduct a small human evaluation of machine translation output using our technique.

## 2.2 RESULTS

We show automatic evaluation results in Table 1. For all languages, we show improvements from introducing pivot pre-finetuning after 50k examples, however, these results seem to plateau with 100k pivot pairs maintaining similar performance. Given this, we can assert that within our experimental setup, pivot pre-finetuning is indeed helpful, boosting performance by 40% (measured by chrF) over a non-finetuned baseline. Despite the lesser quality of pairs in Kiswahili and Kinyarwanda, we find consistent improvements over using French data, suggesting that language similarity is important.

Our human evaluation (Table 2) also reflects results similar to the chrF metric (suggesting that BLEU is somewhat inaccurate for this language pair), training with the pivot language of Kiswahili with both 50k and 100k improves human evaluation.

## 2.3 CONCLUSIONS

In this paper, we have experimented with *pivot-prefinetuning* with the language of Kikamba, a method to enable more data efficient adaptation to low-resource languages, combining rapid NMT adaptation techniques and pre-trained sequence-to-sequence models. Particularly, we find that using related albeit higher resource languages as an intermediate step does help on both automatic and human evaluation. In future work, we look to expand the languages and model settings we consider.

## URM STATEMENT

The authors acknowledge that at least one key author of this work meets the URM criteria of ICLR 2023 Tiny Papers Track.

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

## A  APPENDIX

**Implementation**   We train and evaluate all models within the HuggingFace Transformers (Wolf et al., 2019). We use two NVIDIA RTX 2080Ti GPUs for all training runs.

### A.1  TRAINING DATASETS

We provide further detail on the datasets we use for training below.

- **French:** We use IWSLT 2017 (Sakti & Utiyama, 2017) high-quality training data for English to French.
- **Kikamba:** We use a combination of the FLoRes `dev` portion and the noisily aligned[1] data used in NLLB (NLLB Team et al., 2022).
- **Kiswahili:** We use the noisily aligned NLLB data. For the 1k language pair pivot pre-finetuning training runs, we use the clean FLoRes `dev` data for training.
- **Kinyarwanda:**We use the noisily aligned NLLB data. For the 1k language pair pivot pre-finetuning training runs, we use the clean FLoRes `dev` data for training.

---

[1]https://huggingface.co/datasets/allenai/nllb

