# OpenReview forum: "Pivot Pre-finetuning for Low Resource MT: A Case Study in Kikamba."
_ICLR.cc/2023/TinyPapers — Submitted to Tiny Papers @ ICLR 2023_

### Official Review · Reviewer_ZMT1 · 2023-03-23

**Confidence:** 4

**Summary Of Contributions:**

The paper proposes to use Neubig & Hu (2018)’s rapid NMT approach for English-to-Kikamba Machine Translation, an extremely low-resource language pair.

**Rating:**

Great Start (GS): a submission which meets some of the reviewing criteria but has room for improvement

**Strengths And Weaknesses:**

Strengths: The paper investigates an important and challenging problem in very low-resource MT.

Weaknesses: The paper lacks clarity on the description of the experimental setup and seems hard to reproduce with very less details about the training data.

**Suggested Changes:**

1. It is unclear how the sizes of training data in Table 1 map to the training dataset discussed in A.1 Adding explicit description on what training data is used and how will be helpful.

2. Table 2 conflates the accuracy of translations with their fluency. Maybe using the sQMT framework [1, Section 3.2] for human annotation is what you need.

[1] [Experts, Errors, and Context: A Large-Scale Study of Human Evaluation for Machine Translation](https://arxiv.org/abs/2104.14478)

---

### Official Review · Reviewer_8YjF · 2023-03-28

**Confidence:** 4

**Summary Of Contributions:**

Good paper however reproducible code missing

**Rating:**

Great Start (GS): a submission which meets some of the reviewing criteria but has room for improvement

**Strengths And Weaknesses:**

1. The paper is well written and addresses low resource languages
2. pivot pre-finetuning ideally using a pretrained language model
3. Performance metrics with 3 different levels of resource languages(low,mid,high) along side with human translation which is best way to go
4. Only drawback is reproducible code not provided in the paper

**Suggested Changes:**

Few corrections are as advised on this paper
1. Spellcheck few words are misspelled like the word “ language” misspelled in the first paragraph
2. Always Ensure complete words are in a single line, for instance due to shortage of space word “intermediate” is split into intermedi- in one line & ate in next line
3. link to the code of the research paper is not provided, unable to determine validity of this paper

---

### Author Response · Authors · 2023-05-30
**I  wish to opt-in for archival**

I  wish to opt-in for archival

---

### Meta-Review · Area_Chair_oVqo · 2023-04-05

**Recommendation:** Invite to archive
**Confidence:** 4

**Metareview:**

**Summary**
* The paper proposes to use Neubig & Hu (2018)’s rapid NMT approach for English-to-Kikamba Machine Translation, an extremely low-resource language pair.

**Strengths**
* The paper investigates an important MT problem in low-resource settings.
* Used a rigorous performance metric that includes 3 different levels of resource languages (low, mid, and high) alongside the human translation.

**Weakness**
* The results are hard to reproduce with very less details about the training data, and the code is not shared.


**Summary:**

The paper proposes to use Neubig & Hu (2018)’s rapid NMT approach for English-to-Kikamba Machine Translation, an extremely low-resource language pair. The main strength, the paper investigates an important MT problem in low-resource settings. Main weakness, the results are hard to reproduce with very less details about the training data, and the code is not shared.

**Comments And Feedback To The Authors:**

* This work proposes a challenging method for an important problem. Where more work is needed in this area for low-resourced languages.

* Besides, please include data and code with these kind of works for the community to easily continue working in such problems.

**Reason For Not Giving A Higher Recommendation:**

* The paper discusses an important problem however, in order to check the validity of the work and reproduce the results, the authors should add a link to the data and their code.

**Reason For Not Giving A Lower Recommendation:**

N/A

---

### Decision · Program_Chairs · 2023-04-10

Invite to archive